# A Novel Master-Slave Architecture to Detect COVID-19 in Chest X-ray Image Sequences Using Transfer-Learning Techniques

**DOI:** 10.3390/healthcare10122443

**Published:** 2022-12-03

**Authors:** Abeer Aljohani, Nawaf Alharbe

**Affiliations:** Computer Science Department, Applied College, Taibah University, Madinah 46537, Saudi Arabia

**Keywords:** COVID-19 detection, computer-assisted diagnosis transfer learning, chest X-ray image, image classification

## Abstract

Coronavirus disease, frequently referred to as COVID-19, is a contagious and transmittable disease produced by the SARS-CoV-2 virus. The only solution to tackle this virus and reduce its spread is early diagnosis. Pathogenic laboratory tests such as the polymerase chain reaction (PCR) process take a long time. Also, they regularly produce incorrect results. However, they are still considered the critical standard for detecting the virus. Hence, there is a solid need to evolve computer-assisted diagnosis systems capable of providing quick and low-cost testing in areas where traditional testing procedures are not feasible. This study focuses on COVID-19 detection using X-ray images. The prime objective is to introduce a computer-assisted diagnosis (CAD) system to differentiate COVID-19 from healthy and pneumonia cases using X-ray image sequences. This work utilizes standard transfer-learning techniques for COVID-19 detection. It proposes the master–slave architecture using the most state-of-the-art Densenet201 and Squeezenet1_0 techniques for classifying the COVID-19 virus in chest X-ray image sequences. This paper compares the proposed models with other standard transfer-learning approaches for COVID-19. The performance metrics demonstrate that the proposed approach outperforms standard transfer-learning approaches. This research also fine-tunes hyperparameters and predicts the optimized learning rate to achieve the highest accuracy in the model. After fine-tuning the learning rate, the DenseNet201 model retrieves an accuracy of 83.33%, while the fastest model is SqueezeNet1_0, which retrieves an accuracy of 80%.

## 1. Introduction

The coronavirus (COVID-19) was first discovered in a city called Wuhan in China’s Hubei Province. These types of cases were also reported earlier as severe acute respiratory syndrome (SARS-CoV) as well as Middle East respiratory syndrome (MERS) [1,2]. Initially, symptoms of COVID-19 were noticed in Wuhan, also known as the Hubei Province of China. This virus has been alarming for the health and comfort of the public. Also, it has widely affected the world economy, businesses, trades, and the stock market. This virus that affects the respiratory system of humans has been declared a pandemic globally. The virus has the same symptoms as that of viral pneumonia. These viruses cause mild to moderate respiratory diseases in humans. It is also noteworthy that it has a rising death rate if immediate treatment is not provided. In this virus, severe infection occurs in one or both pulmonary sacs, resulting in shortness of breath. Chest CT is utilized for analyzing as well as diagnosing pneumonia. Chest CT has also been remarkably beneficial in identifying the radiographic outlines of COVID-19 [3].

The World Health Organisation (WHO) [2] has suggested employing contact tracing and large-scale testing to handle the pandemic effectively. However, not all countries have the necessary infrastructure for mass testing. Reverse transcription polymerase chain reaction (RTPCR) is applied to distinguish viruses. Nasopharyngeal swabs are used for examining the presence of the ribonucleic acid (RNA) of SARS-CoV-2. This is a widespread method for the detection of this infection. Furthermore, the results of these tests may take 1–2 days if there are few cases, and it may take several days if there are many cases. During this period, there is a considerable probability of suspected patients who are waiting with slight or no symptoms transmitting the infection heavily.

COVID-19 has come to the notice of the government, healthcare workers, and researchers since it was declared a pandemic in March 2020. In this virus, whenever the patient has symptoms like fever, cough, runny nose, shortness of breath, etc., it indicates that the person might be infected with the coronavirus. Doctors refer those patients to lung X-ray imaging to detect the infection in the lungs. Most coronavirus-infected patients have abnormal lung X-rays, with the common observation that lower lobes of the lungs are affected by the peripheral ground glass opacity [4].

Vaccines and early disease diagnosis are the only cure available today to maintain a specific distance from the infected person. The new omicron variant of the coronavirus, which has caused worldwide outbreaks, is less superior than the earlier variants but more infectious. RT-PCR tests do detect the coronavirus. However, a limited number of test kits are available, and even then it is time-consuming. Thus, doctors prefer medical imaging technology like X-ray and chest computed tomography (CT) scans for detecting the prominent symptoms of the coronavirus in the images. Thus, machine learning and deep learning technologies are pivotal in analysing images and diagnosing diseases at an early stage. It helps reduce cases as the person can remain in isolation so that the infection does not spread to another person.

Expert radiologists and significant time are required to analyse lung X-rays, which is a challenge in this pandemic. Hence, a computer-assisted diagnosis (CAD) system is needed to assist doctors, especially radiologists, in analysing medical images for diagnosing COVID-19. The computer-assisted diagnosis (CAD) system is contingent on the image quality for promising results. Thus, there is a need for image-processing techniques for pre-processing, conversion, and enhancement, improving image quality. This process is the intermediate and necessary step for a computer-assisted diagnosis (CAD) system, which hinders the performance of automated learning. Therefore, deep learning (DL) techniques are superior for automatically extracting the features from the images without human involvement. Machine learning and deep learning are providing promising results for medical image analysis like breast cancer [5], lung cancer [6,7], brain tumours [8], classification of lung nodules [9,10], and many more.

This paper aims to use transfer-learning techniques by fine-tuning the hyper-parameters to detect coronavirus from chest X-ray images.

The contributions of our research are to explore state-of-the-art transfer learning techniques through literature analysis to propose a novel master–slave architecture. This is achieved by utilizing transfer-learning models like DenseNet201, SqueezeNet1_0, ResNet34, and VGG16 for COVID-19 detection using chest X-ray images. Also, hyperparameters, like the learning rate, are fine-tuned by finding the most optimized learning rate for increasing the model’s accuracy. Finally, the time and accuracy of the transfer-learning models are compared, and the proposed architecture is also compared using state-of-the-art techniques.

## 2. Literature Review

Machine-learning and deep-learning methods are widely utilized for computer vision [11,12], image processing [13], early medical diagnosis [14], and indexing [15]. The literature review considers deep-learning and transfer-learning techniques for COVID-19 detection using several medical image modalities like CT scans, X-rays, etc.

Researchers are working on finding new solutions to detect and diagnose the impact of COVID-19 on the lungs using medical imaging modalities. Many researchers are using transfer learning for the diagnosis of COVID-19. Apostolopoulos and Mpesiana [16] have done a comparative analysis of various transfer-learning models, like MobileNet v2, Xception, VGG19, Inception, and Inception ResNet v2. The dataset utilized in their experimentation includes 100 X-ray images with 50 images of patients with COVID-19 and 50 healthy humans. The highest accuracy achieved for 2-class classification is 98.75% and for 3-class classification is 93.48% using the VGG19 model. Apostolopoulos et al. [17] utilized the pre-trained model MobileNet to classify COVID-19 from 3905 X-ray images into six classes. Also, 455 X-ray images of COVID-19 patients are merged along with it. The pre-processing is performed to resize the images to 200 × 200 pixels, and data augmentation is performed to increase the dataset’s size. Their suggested model achieved 99.1% and 87.66% accuracy for binary and multiclass classification, respectively.

Similar to the above approaches, Loey et al. [18] suggested a generative adversarial network (GAN) for generating several images as there are limited standard chest X-ray image sequences. Correspondingly, transfer learning was utilized for detecting COVID-19. They used a dataset consisting of 307 images for four classes, i.e., bacterial pneumonia, viral pneumonia, COVID-19, and no findings. In their model, 90% of data was used along with validation. Pre-trained models like AlexNet, GoogleNet, and ResNet18 were used to classify the images into four classes. The accuracy achieved was 80.6% for four-class classification, 85.3% for three-class classification, and 100% for two-class classification.

Alternatively, Afshar et al. [19] proposed the following approach to detect COVID-19. The COVID-CAPS, a capsule-based network, was suggested to classify COVID-19 patients from healthy ones. The dataset used consists of X-ray images. Their approach achieves an accuracy of 95.7% along with considerably fewer parameters than the other trained models, while the pre-trained model achieves 98.3% accuracy. COVID-Net, a deep convolutional neural network (CNN), was proposed by Wang et al. [20] to diagnose COVID-19 from chest X-ray images. The authors also introduced the open-access dataset, which consists of 13,975 chest X-ray images. COVID-Net achieved 92.6% in a three-class classification problem.

To optimize the parameters of CNN models, various optimization methods are used. A similar approach is adopted by Goel et al. [21]. The authors proposed a model using optimized convolutional neural networks (OptCoNet) to classify COVID-19 patients among three classes, i.e., normal, pneumonia, and COVID-19. The grey wolf optimizer extracts optimized features and fine-tunes the trained model for higher accuracy. The proposed model consists of an optimized classification component with higher performance than other models, achieving 97.78% accuracy. Likewise, Sahlol et al. proposed a new approach FO-MPA for detecting COVID-19 [22]. The approach consisted of a pre-trained Inception model for feature extraction, while the swarm-based approach marine predators algorithm (MPA) was used for feature selection. The authors used two publicly available datasets for experimentation. The Kaggle datasets consist of X-ray images of patients infected with COVID-19. In their approach, 51K features were extracted from the Inception model. From that, 130 and 86 features were selected by the proposed model for dataset1 and dataset2, respectively. The authors proved that the accuracy is high when these methods are combined. The proposed approach achieved 98.7% and 98.2% for dataset1 and dataset2, respectively.

In contrast to the previous work, few authors focus on modifying the current techniques for better performance. Mor et al. [23] proposed an approach for the enhancement in detecting COVID-19 cases by utilizing a CycleGAN oversampling application. CycleGAN is used to reproduce suitable and applicable synthetic images that solve the problem of the lack of COVID-19 images and provide the solution to the low-quality problem in the images obtained from medical devices. The various experiments were performed using the data augmentation technique on a dataset consisting of chest X-rays and achieved 98.61% accuracy. Arora et al. [24] suggested a stochastic deep-learning approach for detecting COVID-19 infection using X-ray and CT scan images. Their approach accommodated deep representations over the gaussian before emphasizing the discrimination in feature maps. The experimentation was performed for three classes with publicly available datasets, i.e., normal, pneumonia, and COVID-19.

A class imbalance problem is solved for classifying COVID-19 in chest X-ray image sequences by Chamseddine et al. [25]. This approach utilized SMOTE and weighted loss. Feature selection technique and AdaBoost classifier are utilized for COVID-19 prediction by Soui et al. [26]. This approach used the patient’s symptoms for classification. Sentiment analysis of COVID-19 vaccines was conducted by Mansouri et al. [27]. This approach utilized TextBlob and BiLSTM for analysis. COVID-19 can be predicted using both chest X-ray and CT scan sequences. Chest X-ray is preferred as it takes less time and is cheap. This paper utilizes chest X-ray sequencing for COVID-19 detection.

Aside from the inspiring results produced by the various studies and analyses for COVID-19 detection using various medical modalities, a few complications need to be resolved for further studies and analysis. The fundamental problem for any deep neural network is the lack of balanced training data. Very few authors have worked on it to solve this issue by using augmentation techniques. Also, there is a lack of annotated data. Annotation requires expert knowledge, and it is also a time-consuming process. Thus, by solving these problems, the accuracy of the models can be increased.

## 3. Methodology

A widely used subset of machine learning known as deep learning imparts computers to learn through examples as humans naturally do. It is a prime foundation behind applications such as driverless cars, enabling cars to identify signs like go/halt/stop or to differentiate a person from a streetlight. It also helps to regulate speech in various consumer devices like speakers, mobile phones, TVs, tablets, etc. These techniques are receiving serious notice and providing accurate results that were not previously possible.

Deep-learning methods are computer models that learn to perform tasks straight from unstructured data like text, images, and sound, i.e., automatically extracting features. Deep learning models can perform better than other algorithms providing more accurate results. It also exceeds human-level classification in some cases. Here, a massive set of labelled data and architectures of the many-layered neural network are utilized for training the models.

The proposed master–slave architecture is shown in Figure 1. The architecture is divided into two modules named master and slave. The master module is responsible for pre-processing the dataset and training the transfer-learning models. The trained model’s layers are made static or frozen. Although the slave module will learn from the master module by utilizing the freeze layers, only the last few layers will be trained again to optimize the parameters. Thus, the slave module is responsible for optimization. A detailed description of the steps performed in the proposed approach is described in Algorithm 1.
**Algorithm 1: Proposed Master-Slave Algorithm for detecting COVID-19**.
Assume *i* = chest X-ray image, p = pre-processingStep 1: Read(*i*)Step 2: Perform *p(i)*2.1 Resize (*i*) to 256 × 2562.2 Normalize pixel values (*i*)2.3 Data samplingStep 3: Feature extraction (pretrained models: DarkNet, VGG16, ResNet, SqueezeNet, DenseNet)Step 4: Optimization (freeze layers, learning rate by using optimization technique)4.1 Finding the best learning rate (lr)4.2 Retraining the model concerning lr4.3 Data samplingStep 5: Performance evaluationStep 6: Comparison (state-of-the-art recent technique) 

Below is a detailed description of the master and slave modules and sub-modules.

### 3.1. Dataset

In our research, the dataset utilized for the study consists of X-ray images prepared by integrating images from two different data sources that were used to detect COVID-19 infection. The first dataset comprises COVID-19 X-ray images developed by Cohen et al. [28], including images retrieved from several open-access resources. Also, this source regularly updates images shared by several researchers from different regions. This dataset currently contains 127 X-ray images diagnosed with COVID-19. This dataset consists of 43 female and 82 male patients diagnosed as COVID-19 positive. The second dataset incorporates chest X-rays by Wang et al. [29] that contains normal and pneumonia images. These data sources were integrated and utilized by Ozturk et al. [30,31]. They utilized 500 images of healthy lungs and 500 with pneumonia on frontal chest X-ray images. The images were selected randomly. This was done to prevent the challenge of unbalanced data. In our approach, 100 images from each group include COVID-19, no findings, and pneumonia. A few images of the dataset are shown in Figure 2.

### 3.2. Data Pre-Processing

The images are pre-processed to resize them to 256 × 256 to make all the images the same shape and normalized. In this approach, the utilized dataset is separated into training data and validation data. Overall, 80% of the images are used for training, while 20% are for validation.

### 3.3. Transfer Learning

Deep learning is extensively and effectively used in various domains where there is a need to extract hidden patterns from historical data to predict future results [32]. ML techniques are typically categorized by training and testing datasets containing features and the target variable. There is a high fluctuation in results based on the distribution of the datasets [33]. Hence, there is a requirement for a vast dataset. In many cases, it is impossible to find sizeable training data that matches the feature space giving high accuracy. Additionally, it is challenging to make accurate predictions regarding the data distribution characteristics of test data. As a result, there is high demand for sophisticated learners that can learn more efficiently from small data and achieve high performance. It is possible by transferring the domain knowledge of one task to another related task. It is a fundamental principle for applying transfer learning to various fields.

Transfer learning is based on real-world scenarios. For example, if someone knows how to ride a bike, they can ride a car using the same knowledge. Humans transfer knowledge gained from one domain to another to learn things fast and more efficiently. Similarly, transfer learning uses the previously learned models on some datasets to map the gained knowledge using new data. However, there is a difference between typical ML and transfer learning. In typical ML, algorithms are used to train the model from the start without retaining the knowledge gained from previous training. In contrast, transfer learning retains the previously gained knowledge to train with the new domain or data, as shown in Figure 3.

The various standard pre-trained models for classifying medical images are discussed below.

Consider an image i_0_ that is given as an input to the transfer-learning model, i.e., in the convolutional network. Suppose there are L layers that represent information that is non-linear T_ℓ_ (·), where ℓ is the index of the layer T_ℓ_(·). T_ℓ_ (·) is a compound function that consists of various types of procedures like convolution (Conv), dropout, batch normalization (BN) [34], activation functions like rectified linear units (ReLU) [35], various types of pooling [36] like max pooling, min pooling, average pooling, etc. The output of the *ℓ^th^* layer is *i_ℓ_*.

#### 3.3.1. ResNets

The typical convolutional neural networks, for each ℓ^th^ layer, connect the input with the output layer *(ℓ+1)^th^* layer [37], thus making the transition from i_ℓ_ to *T_ℓ_ (i_ℓ_*_−1_*)*. ResNets [38] are famous for solving the degradation problem by introducing the skip connections that sidestep the non-linear transformations as given in Equation (1).
I_ℓ_ = T_ℓ_ (i_ℓ−1_) + i_ℓ−1._(1)

The advantages of using ResNets are that the gradient is not diminished and can be passed to later layers through the identity function. However, the output of T_l_ is added with an identity function that can delay the information flow in the network.

#### 3.3.2. DenseNet

Direct connections are also introduced from the layers to all the succeeding layers in ResNets to combat the issue of information delay [39]. These connections help to improve the information flow in the network. As a result, the lth layer’s feature maps are taken into account by all of the layers (*i*_0_, *i*_1_, *i*_2_…., *i_ℓ_*_−1_) that came before it as input:i_l_ = T_l_ ([i_0_, i_1_,…, i_ℓ−1_])(2)
where [*i*_0_, *i*_1_, *i*_2_…., *i_ℓ_*_−1_] depicts the combination of the generated feature maps through layers 0,1,2,3—*ℓ*−1. As this network is densely connected, it is known as DenseNet.

#### 3.3.3. DarkNet

DarkNet-19 is the standard model that is used to detect objects. It is designed for classifying objects in a real-time environment which is widely known as “You Only Look Once” (YOLO) [40]. This architecture is utilized for medical imaging because of its successful architecture. Hence, inspired by DarkNet, DarkCovidNet [30] is proposed, which is an optimized model with few layers and filters in contrast to the original DarkNet.

#### 3.3.4. VGG16

VGGNet [41] is designed by Oxford University and is trained on the dataset of ImageNET. The architecture of VGG16 uses 16 weight layers containing 13 convolutional layers having a 3 × 3 filter size and three fully connected layers with a 3 × 3 filter size. AlexNet, as well as VGG16, have the same configurations. The convolutional layers are placed in five groups. In contrast, the max-pooling layer is followed after each group. The padding and stride are kept to 1 pixel for all the convolutional layers. The disadvantage of using the VGG16 is its significant architecture consisting of 160M parameters.

#### 3.3.5. SqueezeNet

SqueezeNet [42] is trained on the ImageNet dataset achieving AlexNet-level accuracy but with smaller network architecture. The larger architectures of convolutional deep neural networks create a bottleneck regarding resource utilization and time consumption. The advantages of using the smaller networks with similarly accurate results are distributed learning and less communication across the servers, reducing the training time. Fewer parameters in the network result in using the lesser band to export the model from the cloud for transfer learning, and smaller networks are feasible for being deployed with fewer hardware requirements like limited memory, etc. Consequently, SqueezeNet was proposed in 2016 with a small CNN architecture having 50× fewer parameters to offer all these advantages. Furthermore, while compressing the model by using compression techniques, SqueezeNet is compressed to less than 0.5 MB.

### 3.4. Performance Measure

The proposed method is evaluated using standard parameters like accuracy and cross-entropy loss.

#### 3.4.1. Accuracy

Accuracy is measured as the ratio of the total count of correctly classified instances and the total count of instances in the dataset.
(3)$$Accuracy=\frac{Tp+Tn}{Total\;Number\;of\;instances}$$
where *Tp* = true positive and *Tn* = true negative.

#### 3.4.2. Cross-Entropy Loss or Logloss

Binary cross entropy is the negative average of the logarithm of correctly predicted probabilities. It compares each predicted probability to the actual label output for binary classification. The score is based on the difference from the actual expected value to penalize the probabilities [43].
(4)$$Log\;loss=-\frac{1}{N}\sum_{i=1}^{N}\sum_{j=1}^{L}y_{ij}log\left(p_{ij}\right)$$
where *N* depicts the total instances, *L* represents class labels, *y_ij_* represents the actual value of the *i*th instance in the *j*th class, and *p**_ij_* represents the predicted value of the *i*th instance in the *j*th class.

### 3.5. Training and Classification

The proposed architecture uses pre-trained models like DarkNet, VGG16, ResNet34, SqueezeNet1_0, and DenseNet201. The initial layers of the model are frozen while the learning rate is fine-tuned to optimize the model for higher accuracy. The last few layers are retrained to obtain the optimized model. The advantage of this approach is it does not increase the complexity of the standard DenseNet201 model.

### 3.6. Test and Evaluation

The proposed model is evaluated and tested using standard performance metrics like accuracy and cross-entropy loss.

## 4. Results and Discussion

This section showcases experiments performed for multiclass classification using the proposed model DenseNet201 to differentiate COVID-19 X-ray images from those of normal and pneumonia cases. The proposed approach is evaluated using the other four models, i.e., DarkNet, ResNet34, SqueezeNet1_0, and VGG16. It is performed using python, especially utilizing fastai, a deep learning library. The various parameters for all four models are given in Table 1. All the models are trained for ten epochs.

Table 2 represents the models’ training loss, validation loss, and accuracy after 10 epochs. From the table, DenseNet201 outperforms other models in classifying COVID-19 images, achieving 85.00% accuracy.

Figure 4, Figure 5, Figure 6, Figure 7 and Figure 8 depict the assessment of training as well as validation losses for DarkNet, ResNet34, VGG16, SqueezeNet1_0, and DenseNet201, respectively.

From the above Figure 4, Figure 5, Figure 6, Figure 7 and Figure 8, it is clear that, initially, the training and validation losses are high, then, as the epochs increase, they eventually decrease. Mostly, the training and validation losses stabilize for all the models trained after 65 batches. After 10 epochs, i.e., 70 batches, there is a minor change in training and validation losses for every model. It also depicts that the losses of the DenseNet201 model are less compared to the other models, proving it to be better than the other four models.

Figure 9, Figure 10, Figure 11 and Figure 12 illustrate the comparison of accuracy, training loss, validation loss, and time across the 10 epochs, respectively.

Figure 9, Figure 10, Figure 11 and Figure 12 depict that the accuracy of all the models has increased considerably after the eighth epoch and stabilizes at the ninth epoch. Figure 9, Figure 10 and Figure 11 also showcase that DenseNet201 outperforms other models achieving an accuracy of 83%, while DenseNet201 and SqueezeNet1_0 have minimum losses in contrast to the other four models. Moreover, the fastest models among the five are DarkNet and SqueezeNet, as depicted in Figure 11.

The transfer-learning models are fine-tuned to find the best learning rate. Figure 13, Figure 14, Figure 15 and Figure 16 depict the learning rate concerning the loss for ResNet34, VGG16, SqueezeNet1_0, and DenseNet201, respectively. The red dot indicates the best learning rate for the respective model that can give the highest accuracy. Thus, for all the methods, the learning rate is selected to be nearer to the red dot, as shown in Figure 13, Figure 14, Figure 15 and Figure 16.

The best learning rate and the accuracy obtained after optimizing the learning rate parameter are showcased in Table 3. It also compares the accuracy before and after fine-tuning the hyperparameter learning rate.

Thus, Table 3 shows that the accuracy of the models ResNet34, VGG16, and SqueezeNet is increased while the accuracy of the DenseNet201 model is the same. Nevertheless, in terms of accuracy, it performs better than any of the other four models.

The results of the DenseNet201 and SqueezeNet1_0 models after fine-tuning are showcased in Figure 17 and Figure 18. In Figure 17 and Figure 18, the first represents the ground truth value, and the second represents the predicted value for the given image below.

The research proposes a novel master–slave methodology that utilizes the transfer-learning approach for detecting COVID-19 by analyzing chest X-ray images. Among the models used, the DenseNet201 model outperforms, giving an accuracy of 83.33%, while the fastest model is SqueezeNet1_0, achieving an accuracy of 80% after fine-tuning the learning rate. DenseNet201 performs well on the small dataset, extracting low and high-complexity-level features. Hence, the time required by the model is high. In contrast to the DenseNet201 model, SqueezeNet1_0 is 20 times faster, giving comparable accuracy as it has lightweight architecture with significantly fewer trainable parameters. The training process for deep-learning models is time-consuming. Therefore, high-performance graphical processing units (GPUs) can achieve faster results.

## 5. Conclusions and Future Work

The new coronavirus disease 2019 (COVID-19) has caused a worldwide pandemic. There is an increased demand for COVID-19 testing and diagnosis due to the high number of cases. Cases of this virus are increasing daily, putting a significant strain on our social, emotional, physical, and mental health. Moreover, early detection of COVID-19 cases is a crucial and beneficial step in the virus’ prevention. One of the primary reasons for the rapid spread of COVID-19 cases is the traditional time-consuming testing process. A chest X-ray or other imaging tool could be used instead to expedite the testing process and stop the spread of the virus. As a result, this study aims to create a computer-assisted diagnosis (CAD) system to classify samples of COVID-19 from a chest X-ray dataset that includes pneumonia, COVID-19, and healthy cases. The study focuses on analyzing X-ray images for early COVID-19 detection. With an accuracy of 83.33%, the proposed master–slave approach with DenseNet201 outperforms other models. The research can be expanded in the future to improve model accuracy by using data augmentation techniques to increase the dataset size. Furthermore, GPUs can be used to train models quickly.

## Figures and Tables

**Figure 1 healthcare-10-02443-f001:**
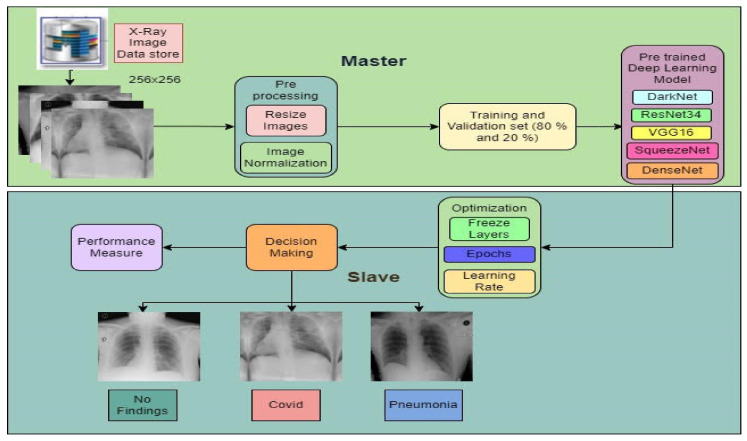
Proposed architecture.

**Figure 2 healthcare-10-02443-f002:**
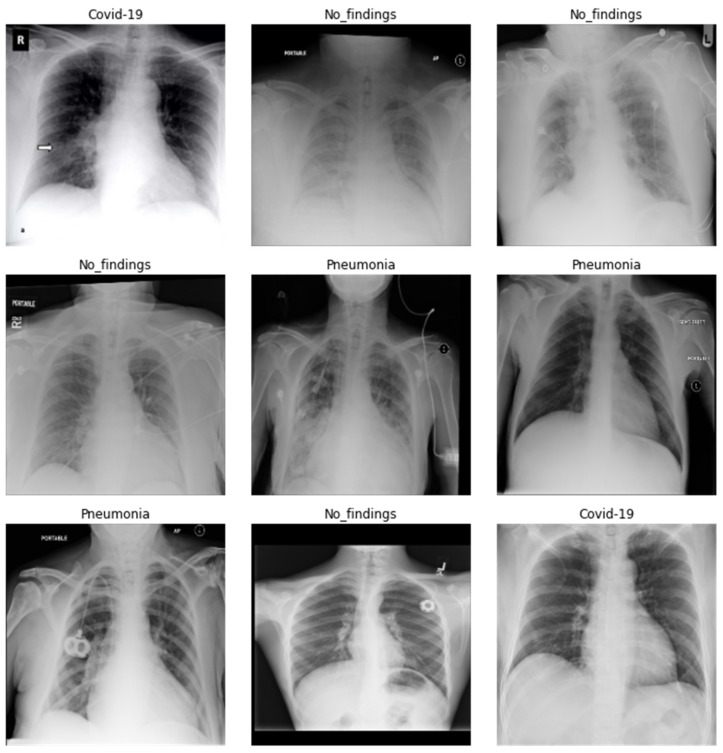
Dataset images.

**Figure 3 healthcare-10-02443-f003:**
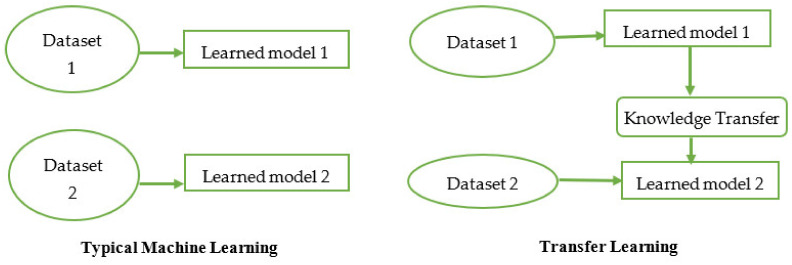
Comparison between typical machine learning and transfer learning.

**Figure 4 healthcare-10-02443-f004:**
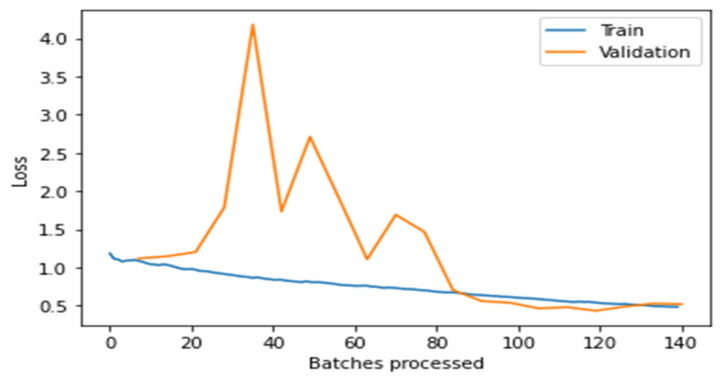
Comparison of training loss and validation loss for DarkNet.

**Figure 5 healthcare-10-02443-f005:**
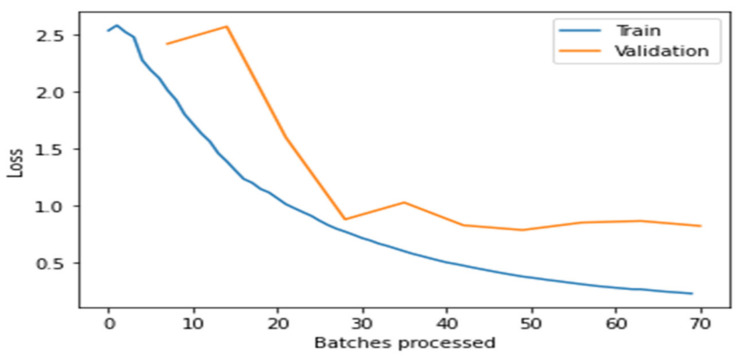
Assessment of training and validation loss for ResNet34.

**Figure 6 healthcare-10-02443-f006:**
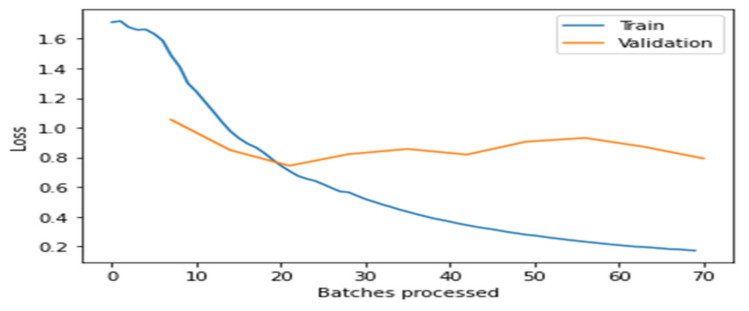
Comparison of training and validation loss for VGG16.

**Figure 7 healthcare-10-02443-f007:**
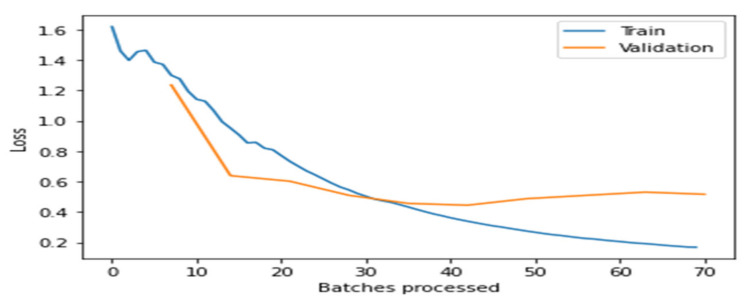
Comparison of training and validation loss for SqueezeNet1_0.

**Figure 8 healthcare-10-02443-f008:**
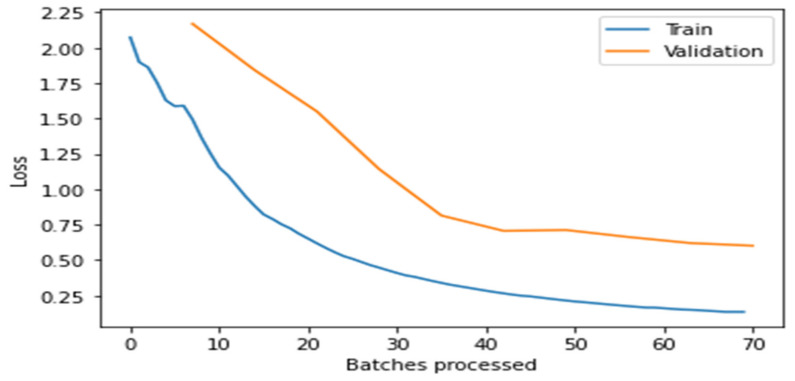
Comparison of training and validation loss for DenseNet201.

**Figure 9 healthcare-10-02443-f009:**
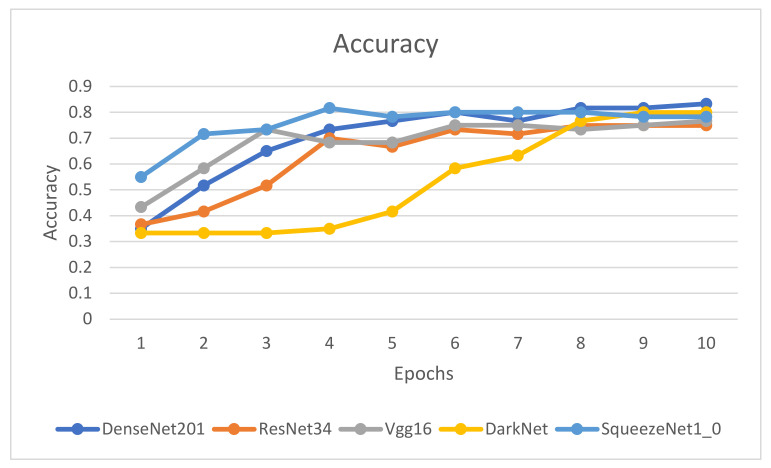
Comparison of accuracy for epochs.

**Figure 10 healthcare-10-02443-f010:**
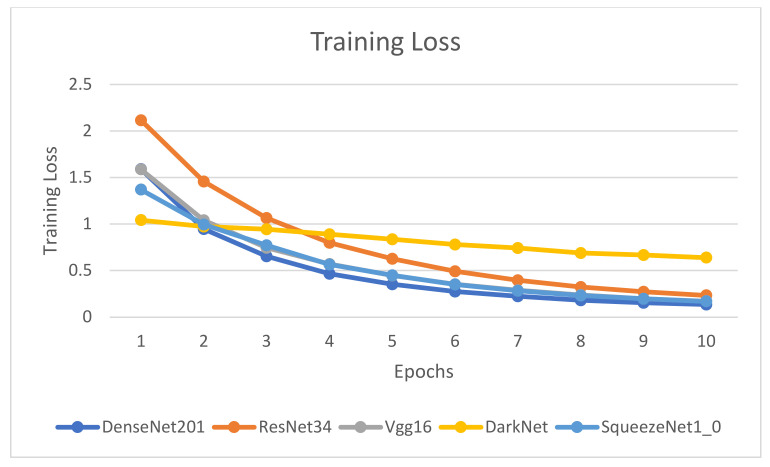
Comparison of training loss for epochs.

**Figure 11 healthcare-10-02443-f011:**
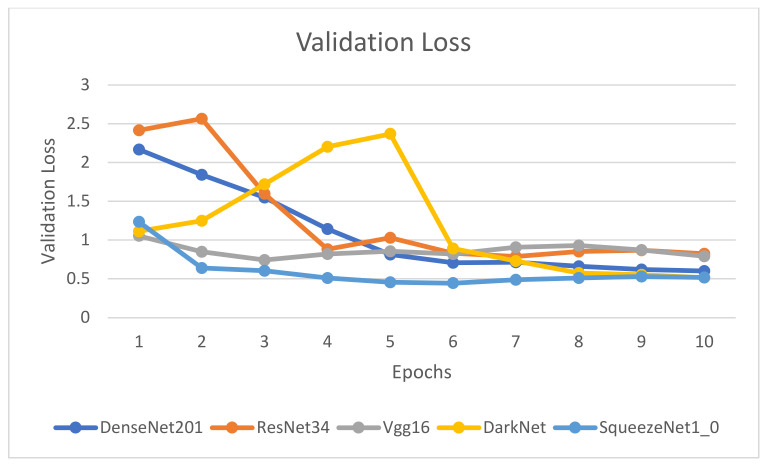
Comparison of validation loss for epochs.

**Figure 12 healthcare-10-02443-f012:**
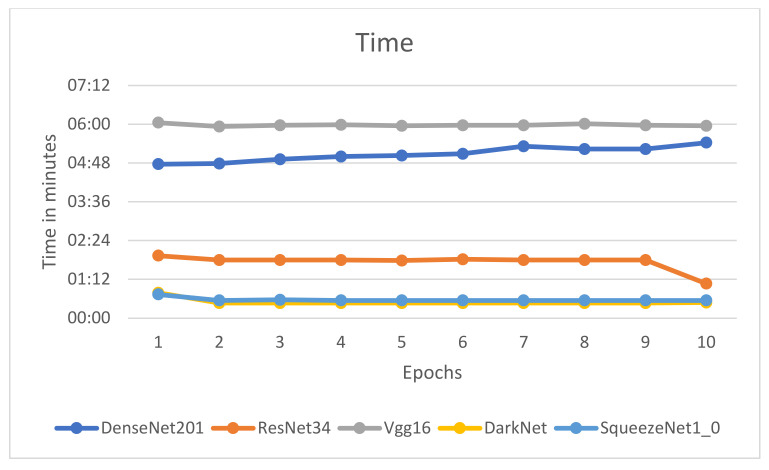
Comparison of time for epochs.

**Figure 13 healthcare-10-02443-f013:**
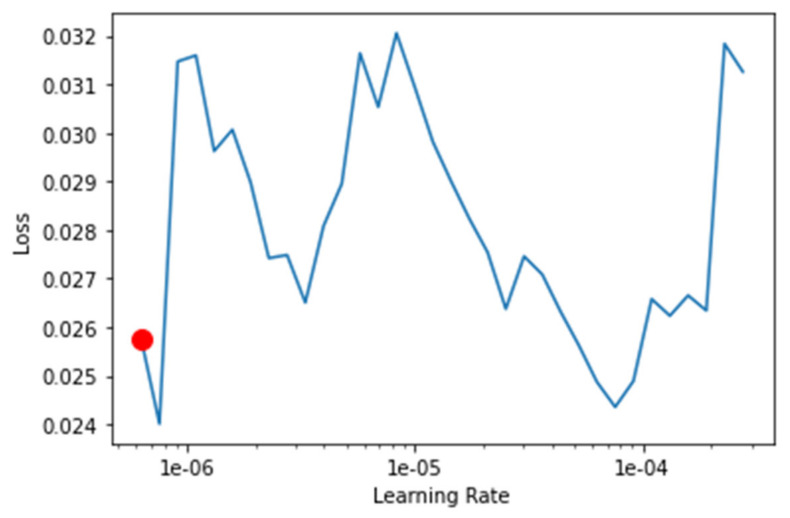
Graph of the learning rate for the loss for ResNet34.

**Figure 14 healthcare-10-02443-f014:**
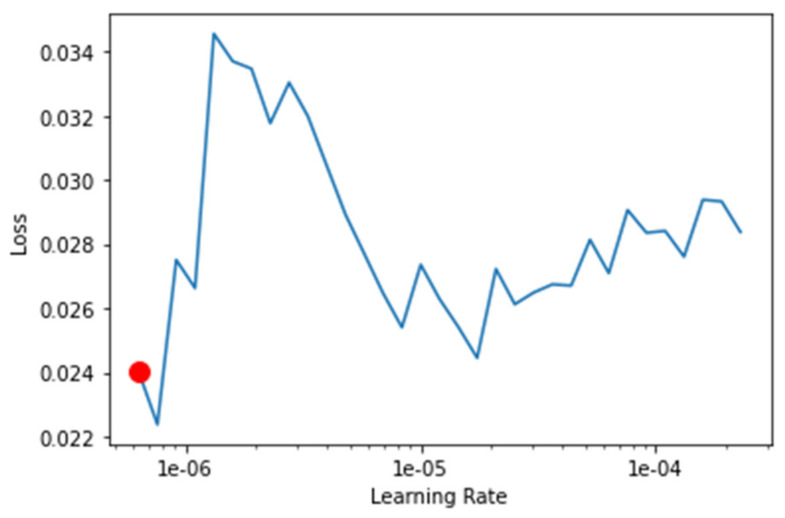
Graph of the learning rate for the loss for VGG16.

**Figure 15 healthcare-10-02443-f015:**
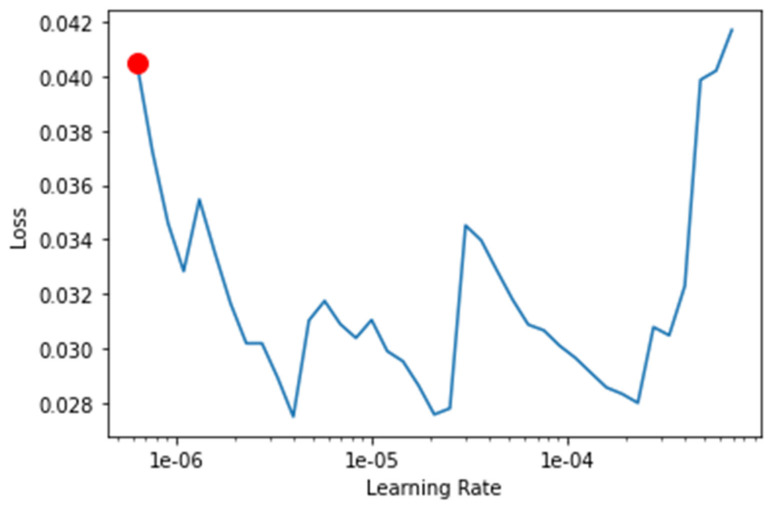
Graph of the learning rate for the loss for SqueezeNet1_0.

**Figure 16 healthcare-10-02443-f016:**
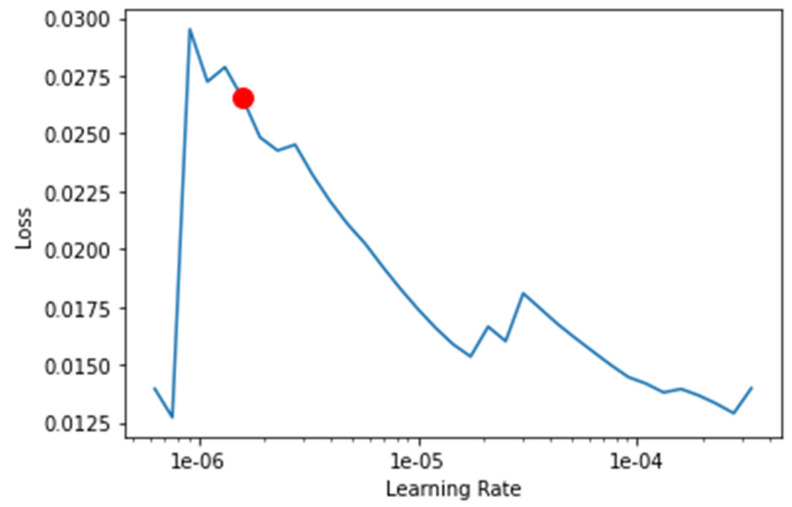
Graph of the learning rate for the loss for DenseNet201.

**Figure 17 healthcare-10-02443-f017:**
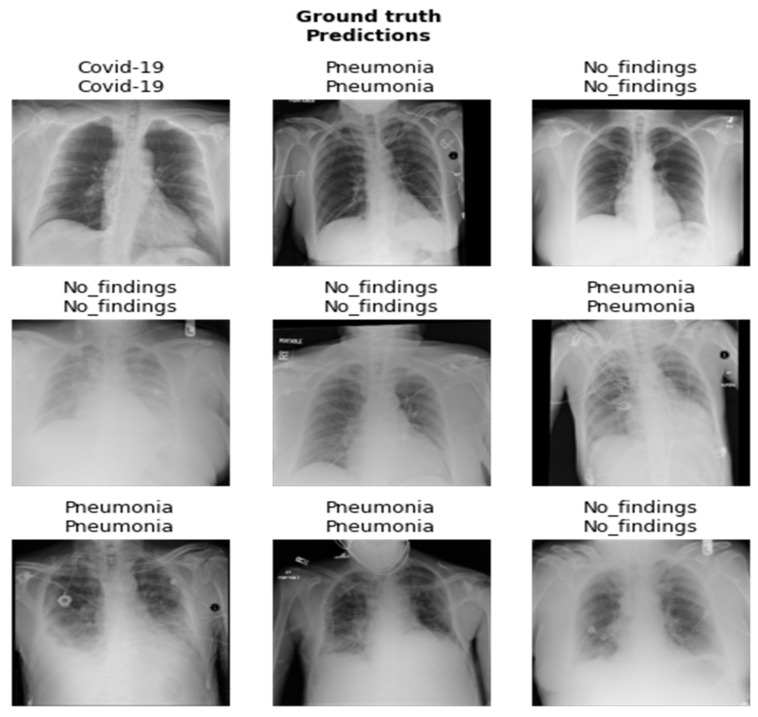
Results of DenseNet201 model after fine-tuning.

**Figure 18 healthcare-10-02443-f018:**
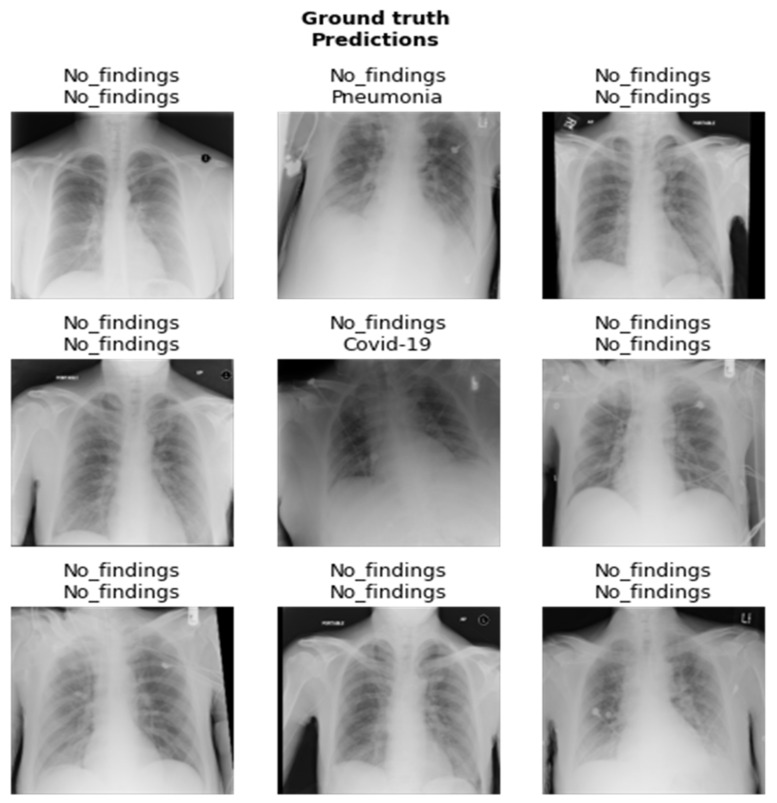
Results of SqueezeNet1_0 model after fine-tuning.

**Table 1 healthcare-10-02443-t001:** Parameters used for the proposed methodology.

Model	Depth	Number of Parameters	Optimizer
DarkNet	56	1,167,586	Adam
ResNet34	34	21,814,083	Adam
VGG16	16	15,252,547	Adam
SqueezeNet1_0	14	1,264,835	Adam
DenseNet201	201	20,069,763	Adam

**Table 2 healthcare-10-02443-t002:** Results of all the models after 10 epochs.

Model	Train Loss	Valid loss	Accuracy
DarkNet	0.638718	0.515339	0.800000
ResNet34	0.233139	0.823872	0.750000
VGG16	0.171060	0.792268	0.766667
SqueezeNet1_0	0.168784	0.516652	0.783333
**DenseNet201**	**0.135499**	**0.601770**	**0.833333**

**Table 3 healthcare-10-02443-t003:** Fine-tuned learning rate and accuracy.

Model	Initial Learning Rate	Learning Rate after Fine-Tuning	Accuracy before Fine-Tuning	Accuracy after Fine-Tuning
DarkNet [22]	3e−3	-	0.800000	-
ResNet34 [44]	Default	1e−06, 1e−07	0.750000	0.783333
VGG16 [45]	Default	1e−07, 1e−06	0.766667	0.800000
SqueezeNet1_0	Default	1e−06, 1e−07	0.783333	0.800000
DenseNet201	Default	1.58e−06	**0.833333**	**0.833333**

## Data Availability

Data sharing does not apply to this article as no datasets were generated during the current study.

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
