# Peer review of "A Novel Master-Slave Architecture to Detect COVID-19 in Chest X-ray Image Sequences Using Transfer-Learning Techniques"

_healthcare, 2022, doi:10.3390/healthcare10122443_

Round 1

Reviewer 1 Report

The paper proposes to use the pre-trained DenseNet201 and SqueezeNet1_0 architecture for COVID-19 detection from chest X-Ray image classification. The papers has serious lack in novelty. I have few critical concerns regarding the paper.

1. The name master-slave architecture is not suitable as the proposed approach just finetunes the parameters of last few layers in the pre-trained CNN architectures.

2. The approach is said to have used only 100 images of COVID-19 patients. However there are also other datasets available like the one available in "Gupta R. COVID19 classifier dataset. 2020. https://www.kaggle.com/rgaltro/newdataset." How did the approach proposed here work for this dataset.

3. Many pre-trained CNN architecture has achieved classification accuracy of 99% for the dataset used in this paper (refer: FractalCovNet architecture for COVID-19 Chest X-ray image Classification and CT-scan image Segmentation,
Biocybernetics and Biomedical Engineering"). The architecture proposed in this approach achieves only 83.33% accuracy. How is this a better approach. This paper is missing the comparison with other existing approaches.

4. The dataset is spit as 80% for training 20% for validation. Then what about test split? Is the results listed are only for validation dataset? Then how that can be validated?

Author Response

kindly see the attached file 

Reviewer 2 Report

The article utilizes transfer learning to classify x-ray images for the COVID-19 diagnosis. The authors then used five pre-trained models with fine-tuning.  However, I found that the paper’s contribution is significantly weak. Furthermore, some points need to be considered in this study. Regarding the comments given below, this paper needs some significant revisions both from scientific and structural aspects. The changes proposed should be considered in order to improve the quality of this paper. 

1-     The paper suffers from major language problems. The language should be further improved and thoroughly check the manuscript to omit grammatical errors.

2-     Introduction:

-        The authors talked about CT and X-ray images in two separate places in the introduction section. For more clarity about the use of imaging modality for COVID-19 detection,  it will be better to put them in one paragraph. The medical significance of chest X-rays over chest CT scans needs to be mentioned.

-        The paragraph about the use of deep learning techniques should be improved. The author should talk more about DL, Transfer learning, and the impact of DL in the Medical Image Analysis (MIA) field. It is better to use “a computer aid diagnosis system (CAD)” than using an “automated system”.

-        Authors should provide the research objectives and contributions of the proposed work in more detail.

-        I advise authors to cite and to follow the following article:

Ekram Chamseddine, Nesrine Mansouri, Makram Soui, Mourad Abed. “Handling class
imbalance in COVID-19 chest X-ray images classification: Using SMOTE and weighted loss
”, Applied Soft Computing, 2022, 109588, ISSN 1568-4946,
https://doi.org/10.1016/j.asoc.2022.109588.

.Soui, M., Mansouri, N., Alhamad, R., Kessentini, M., & Ghedira, K. (2021). NSGA-II as feature selection technique and AdaBoost classifier for COVID-19 prediction using patient’s symptoms. Nonlinear Dynamics, 1-23. https://www.ncbi.nlm.nih.gov/pmc/articles/PMC8129611/

Mansouri, N., Soui, M, Alhasan I, Abed Mourad, TextBlob and BiLSTM for Sentiment analysis toward COVID-19 vaccines, 7th International Conference on Data Science and Machine Learning Applications (CDMA 2022).

3-     Literature review:

-        Mention the papers which are more relevant to the field of work. Provide contributions and limitations of each work.

4-     Methodology:

-        The readability and presentation of the section should be further improved. This section needs more highlights to show what your contribution is and how it can be used to affect the dataset.

-        The introduction paragraph about the use of DL is not coherent. Better to use examples related to the MIA field rather than autonomous vehicle. I suggest a separate subsection to present DL, illustrate CNN and how it works, and its impact on MIA. (3.1. Deep learning).

-        Transfer learning should be in a separate subsection (3.2. Transfer learning). This paragraph needs to be rewritten. It should summarize the idea of Transfer learning. Redraw Figure 3. It is better to add a figure for the architecture of transfer learning. Rewrite a simple and clear description to each architecture (DenseNet201, SqueezeNet1, ResNet34, VGG16, and DarkNet) in a separate sub-subsection (i.e. 3.2.1 DenseNet201). Explain why the authors selected these CNNs to be trained not others.

-        Rename the “Architecture description” subsection to “3.3 Proposed method”

-        The pre-processing and data split subsections should be merged into one subsection called “3.3.1. Data pre-processing”.

-        Add a new subsection call it “3.3.2 Training and classification” in which the author should provide and explain all the details about the training, optimization, and classification steps.

-        Add a new subsection call it “3.3.3 Test and evaluation” in which the author should provide and explain all the details about testing and evaluation steps.

5-     Add a new section “4- Experiment setup” in which the author first describes the used dataset and second the evaluation criteria.

-        In the “4.1 Dataset”, the author should clarify the number of samples in each class in the final dataset. Did the other consider taking equal samples number for each class in order to handle the class imbalance issue? please clarify. In “Figure 2. Dataset images”, one row is enough to show samples from the dataset.

-        Add a subsection “4.2 Evaluation Criteria” to illustrate the evaluation criteria (Performance measure). Some more performance evaluation parameters need to be calculated including; Sensitivity, Specificity, F1-Score, and AUC.

6-     In the “Experiment and Results” section:

-        Merge sections 4 and 5. Rename this section to “5. Results and discussion”

-        I understand that the author only fine-tuned the learning rate.  What about the other hyperparameters?

-        The contributions and results should be discussed more.

-        Not all figures and tables are discussed.

-        Detailed layer-wise architecture of the finest performing model needs to be discussed.

-        Discuss the explanations for the provided results.

7-     Discuss the computation complexity of the proposed work.

8-     The conclusion and future work section should be rewritten.

Author Response

kindly see the attached file 

Reviewer 3 Report

Notes:

1. The inscriptions in figures 2, 17, 18 are not clear.

2. In figure 3, not all text is readable on the structural elements of the figure.

3. Pay attention to the structure of section 3. There is a subsection 3.1, and in it 3.1.1 ... 3.1.5. No subsection 3.2. May change the structure of section 3.

4. Lines 120, 123, 413, the word Covid-19 with a small capital letter. Is it necessary?

5. Check formulas (1), (2), introduction and explanations to them for the correct use of fonts (italics?).

6. Lines 14, 111, 329, 413 X-ray is typed with a small capital letter.

7. Line 189. Algorithm 1. Is the number required?

8. Figures 4-8, 13-16 the inscriptions on the figures are fuzzy, including the axes. The style of the drawings is different from the styles of other drawings in the article.

9. The correctness of the format of numbers on the x-axis in fig. 13-16 and in the third column of Table 3 is questionable. Maybe write in this format 1.723´10-32.

10. Check the correctness of the design of the list of references, for example, 473-474, 479-480, etc. (font of the title of journals and other sources).

I recommend carefully proofreading the text of the article in order to eliminate technical errors.

Author Response

kindly see the attached file 

Round 2

Reviewer 1 Report

The manuscript lacks novelty in the proposed approach.

Author Response

kindly check the attached file

Reviewer 2 Report

all the comments are taken in consideration in this improved version 

Author Response

kindly check the attached file
